

# Dome effect of black carbon and its key influencing factors: A one-dimensional modelling study

Zilin Wang[1,2], Xin Huang[1,2]*, Aijun Ding[1,2]

[1]Joint International Research Laboratory of Atmospheric and Earth System Sciences, School of Atmospheric Sciences, Nanjing University, Nanjing, 210023, China
[2]Jiangsu Provincal Collorative Innovation Center of Climate Change, Nanjing, 210023, China

*Correspondence to*: xinhuang@nju.edu.cn

**Abstract**

Black carbon (BC) has been identified to play a critical role in aerosol-planet boundary layer (PBL) interaction and further deterioration of near-surface air pollution in megacities, which has been named as its "dome effect". However, the impacts of key factors that influence this effect, such as the vertical distribution and aging processes of BC, and also the underlying land
surface, have not been quantitatively explored yet. Here, based on available in-situ measurements of meteorology and atmospheric aerosols together with the meteorology-chemistry online coupled model, WRF-Chem, we conduct a set of parallel simulations to quantify the roles of these factors in influencing the BC's dome effect and surface haze pollution, and discuss the main implications of the results to air pollution mitigation in China. We found that the impact of BC on PBL is very sensitive to the altitude of aerosol layer. The upper level BC, especially those near the capping inversion, is more essential in
suppressing the PBL height and weakening the turbulence mixing. The dome effect of BC tends to be significantly intensified as BC aerosol mixed with scattering aerosols during winter haze events, resulting in a decrease of PBL height by more than 25%. In addition, the dome effect is more substantial (up to 15%) in rural areas than that in the urban areas with the same BC loading, indicating an unexpected regional impact of such kind of effect to air quality in countryside. This study suggests that China's regional air pollution would greatly benefit from BC emission reductions, especially those from the elevated sources
from the chimneys and also the domestic combustions in rural areas, through weakening the aerosol-boundary layer interactions that triggered by BC.

**Key words:** black carbon, aerosol-boundary layer interaction, regional haze pollution, WRF-Chem



# 1 Introduction

Air pollution, particularly haze pollution, has been one of the key environmental challenges to China, especially in developed regions like the northern and eastern China (Wang et al., 2017). The haze pollution in these regions is generally characterized as extremely low visibility and dramatically rising surface aerosol concentration (Cai et al., 2017;Zhao et al., 2011). For example, in January 2013, a long-lasting episode of severe haze occurred in central and eastern China with the maximum $PM_{2.5}$ (particles with dynamic diameter less than 2.5 µm) mass concentration in Beijing reaching up to 680 µg m$^{-3}$ (Wang et al., 2014a;Wang et al., 2014c;Wang et al., 2014d). In addition to deteriorating air quality in megacities like Beijing, large-scale regional haze pollution also covers rural, suburban areas (Xu et al., 2011;Chen and Wang, 2015). During serious haze pollution in that month, the thick haze engulfed 1.4 million km$^2$ land area, affecting up to 800 million people in 17 provinces. Such severe and aggravating regional haze pollution has triggered extensive public panic due to $PM_{2.5}$ associated adverse health effect (e.g. cardiovascular and respiratory diseases) (Kim et al., 2015;Mauderly and Chow, 2008 ;Gao et al., 2015a;Pope et al., 2002). Consequently, air pollution mitigation has been one of the top priority for China's central and local governments. However, even though a series of control measures, such as the Action Plan on Prevention and Control of Air Pollution, have been carried out in order to reduce emissions and mitigate fine particle pollution (Zhang et al., 2016), the frequencies of severe pollution events still keep increasing in recent years and the intensity of pollution episode has not shown significant decrease yet (Ding and Liu, 2013;Niu et al., 2010;Cai et al., 2017).

During hazy days, concentration of BC, one of the most important aerosol components from both environmental and climate perspectives, could exceed 20 µg m$^{-3}$ in China's megacities (Yang et al., 2007;Sun et al., 2014), far more than those in other regions across the world. Such a high level of BC concentration in China is primarily attributed to intensive residential combustions and coal-dominant energy structure (Qin and Xie, 2011;Zhang et al., 2009b). Featuring high light-absorbing efficiency, BC would exert substantial impact on climate change at regional and even global scale (Bond et al., 2013;Menon et al., 2002;Ramanathan and Carmichael, 2008). Meanwhile, BC has been proven to be inextricably linked to short-term changes in public health such as cardiovascular mortality and cardiopulmonary hospital admissions (Eklund et al., 2014;Janssen et al., 2011). A recent study by Ding et al. (2016) has revealed the vital role of BC in enhancing near-surface haze pollution by the combined effects of heating by the light-absorbing BC aerosols in upper-PBL and the reduction of surface heat flux, which substantially suppresses the development of PBL and consequently causes extreme haze pollution episode in China's megacities. Such kind of effect was named as the "dome effect" of BC (Ding et al., 2016). Similar effect has also been found for BC over the Indian Ocean (Wilcox et al., 2016), and dust aerosols in northern and eastern China (Liu et al., 2016;Yang et al., 2016).

In terms of the aerosol-PBL interaction induced by BC, the vertical distribution of the aerosol layer is expected as an important influencing factor since that the upper-level BC would alter the air temperature stratification much more efficiently (Ding et al., 2016). Located in one of the main monsoon regions, East China is subjected to large-scale monsoon circulations, where frequent cyclones, fronts and convections tend to lift near-surface air pollutants to the middle and even upper troposphere



(Ding et al., 2009;Zhang et al., 2009a). In addition, the elevated sources like power plant plumes and biomass burning smoke and subsequent long-range transport of air masses frequently lead to vertical heterogeneity of BC profile (Ding et al., 2013;Huang et al., 2016;Yang et al., 2015;Guinot et al., 2006;Chen et al., 2017b). Thus it is of great importance to understand the sensitivity of the dome effect to vertical distribution of BC in China. On the other hand, during haze events, the coexistence

of concentrated air pollutants and complex physicochemical interactions among them are highly possible to lead to a dramatic increase in secondary aerosols (Huang et al., 2014a;Huang et al., 2014b). Hence, freshly emitted BC has been usually observed to undergo notable aging and hygroscopic growth and got almost internally mixed with scattering secondary aerosols like sulfate during hazy days (Bond et al., 2013;Cui et al., 2016;Huang et al., 2013), thereby remarkably enhancing its light-absorbing properties (Peng et al., 2016;Chen et al., 2017a;Cappa et al., 2012;Yang et al., 2012;Shen et al., 2017). Subsequently,

the dome effect of BC might be significantly strengthened by mixing with scattering aerosols. However, such kind of influence to the dome effect of BC hasn't been quantitatively examined yet. As aforementioned, haze pollution generally occurred at regional scale in East and North China. Previous studies have highlighted the importance of aerosol-PBL interaction in cities (Petaja et al., 2016;Ding et al., 2016;Wang et al., 2014a;Gao et al., 2015b;Cai et al., 2017;Li et al., 2017b). It is noteworthy that more than half of the population live in the rural area with a majority in the plain areas in the North and Central China,

who also has been exposed to fine particulate pollution. The rural areas in East China are usually covered by cropland with different land-surface properties to urban area. The difference of the dome effect over the two regions with distinct land cover categories remains to be further explored. In consequence, this study aims at identifying and quantitatively assessing the dependence of the dome effect on several key factors, including vertical distribution, aging of BC and the different underlying surface land (i.e. urban and rural areas).

Numerical simulation with meteorology-chemistry online coupled models has served as a practicable and effective way to characterize aerosol radiative effect and its impact on PBL evolution (Grell et al., 2005;Baklanov et al., 2014;Yu et al., 2002). To disentangle the impacts of various factors on aerosol-PBL interaction, one-dimensional meteorology-chemistry online coupled model is applied in this study for the purpose of excluding influences from synoptic processes and regional transport, etc. Additionally, one-dimension modelling with high vertical resolution enables better representation of PBL evolution and

also allows flexible initial and boundary conditions. Therefore, in the present work, single column version of WRF-Chem (Weather Research and Forecasting model coupled with Chemistry) driven by available observations is employed to investigate the aerosol-PBL interaction and its influencing parameters. The paper is organized as follows. Section 2 discusses major aspects of model simulations, including WRF-Chem model and its configurations, parameterizations of aerosol properties, initial and boundary meteorological and chemical conditions, and the design of numerical experiments. Dome effect

due to BC and its dependence on different conditions are presented and discussed in section 3. Specifically, discussions in section 3 include the effects of vertical distribution and aging process of BC, as well as different underlying land surface. Main results and the possible implications in future policy of air pollution mitigation are discussed in section 4.



## 2. Data and method

### 2.1 Model configuration

The simulations were conducted with the WRF-Chem version 3.6.1 (Grell et al., 2005) single-column model (SCM). Except for advection, the physical and chemical processes are exactly the same with the three-dimensional version, which is a fully

coupled online meteorology-chemistry model including emission and deposition of pollutants, gaseous and aqueous chemical transformation, aerosol chemistry and dynamics. The WRF-Chem SCM runs on a 3×3 stencil with periodic lateral boundary conditions in both zonal and meridional directions. We used a spatial resolution of 4 km and a vertically stretched sigma coordinate with the model top set at a constant pressure, corresponding to about 6000 m. 100 vertical levels were placed equidistantly with height of 60 m from the ground surface to model top to better resolve the vertical structure of the atmosphere

and ensure identical BC mass loadings.

The parameterization schemes were selected following the work by Ding et al (2016), in which the model configuration showed good performance on boundary layer meteorology. The RRTMG short- and longwave radiation scheme (J. Iacono et al., 2008) were used to couple with aerosol scheme in order to reproduce aerosol-radiation interactions. The YSU non-local K boundary layer scheme (Hong et al., 2006) and the Noah land surface scheme (Tewari et al., 2016) were applied for boundary layer

evolution and land-atmosphere interactions, respectively. For representation of cloud and precipitation processes the Lin microphysics scheme (Lin, 1983) together with the Grell-Deveny cumulus parameterization (Grell and Devenyi, 2002) were employed. CBMZ (Carbon Bond Mechanism version Z) photochemical mechanism combined with MOSAIC (Model for Simulating Aerosol Interactions and Chemistry) aerosol model (Fast et al., 2006;Zaveri and K. Peters, 1999) were applied to represent atmospheric chemistry. Aerosols were assumed to be spherical particles. The size distribution was divided into four

discrete size bins defined by their lower and upper dry particle diameters (0.039-0.156, 0.156-0.625, 0.625-2.5 and 2.5-10.0 μm). In terms of mixing state of multiple aerosol chemical compositions, we presume that aerosols in each bin were internally mixed. The extinction coefficient, single-scattering albedo (SSA) along with asymmetry factor were calculated based on Mie theory using volume averaged refractive index. Model domains and configuration selections are summarized in Table 1.

The WRF-Chem SCM model was initialized with monthly averaged radio-sounding measurements at 12:00 UTC taken in

Beijing (116.28°E, 39.93°N) to represent the typical atmospheric stratification in winter-time. Daily sounding data is archived at: http://weather.uwyo.edu/upperair/sounding.html. The soil temperature profile was taken from the monthly averaged WRF regional simulation results for December 2013. The initial condition of atmospheric pollutants, which is mainly concerned with BC will be given in details in the following section. Vertical mixing of aerosol is switched off to ensure that the concentration and altitude of aerosol layer do not vary with PBL evolution. Each numerical experiment was conducted for the

time period of 72 hours with the first 48 hours as model spin-up time.



## 2.2 Design and analysis of numerical experiments

We conducted a set of multidimensional experiments. Each individual experiment contains hundreds of simulations and can be divided into two groups. They shared exactly the same model settings and configurations except that one with aerosol radiation interaction (ARI) while the other without (noARI). Each group consists of several parallel experiments with various initial conditions of airborne pollutants or surface parameters. To investigate the impact of heterogeneity in vertical distribution of BC on the dome effect, BC plumes with a width of 300 meters was settled at different altitude, ranging from the ground surface to about 2000 m. BC concentration was assumed to be 0~30 μg/m$^3$ according to existing field measurements across China (Zhang et al., 2013;Sun et al., 2004;Zhao et al., 2013a). BC concentrations in the plume was presumed to be a Gaussian distribution in vertical with maximum value that ranges from 0 to 30 μg/m$^3$ in the central axis, the altitude of which also ranges from 150 to 2250 m with an interval of 300 m.

Furthermore, to better understand the difference of the dome effect over different underlying surface ground (i.e., urban and rural areas) and its roles in regional air pollution, we conducted the similar parallel numerical experiments over both urban and rural land surface. The distinct differences in important surface parameters between them are listed in Table 2. It is well known that BC emission is mainly related to fossil fuel combustion in China, hence it is usually co-emitted with other gaseous pollutants like $SO_2$ and $NO_x$ and then mixed with their oxidation products, i.e. sulfate and nitrate, during transportation and aging processes (Wang et al., 2014b;Huang et al., 2013;Cheng et al., 2006). To quantify the impact of mixing with these scattering aerosols, we also performed another two parallel simulations with and without SNA aerosols in the BC plume. The mass ratio of BC to SNA was derived from round-year measurement in Beijing (Zhang et al., 2013).

## 3 Results and discussion

### 3.1 One dimensional modelling of the dome effect of BC

Dome effect of black carbon was first revealed by Ding et al. (2016) based on integrating online coupled regional simulation and corresponding observations during the winter haze event in 2013, when severe $PM_{2.5}$ pollution covered East China with hourly concentrations up to ~ 900 μg m$^{-3}$ and the visibility less than 100 m (Zheng et al., 2015). In this study, it has been proven that BC exert an important role in aggravating haze pollution via aerosol-PBL interaction. To clearly demonstrate BC induced aerosol-PBL interaction, we selected one typical episode during 23$^{rd}$-24$^{th}$ December, 2013 in Beijing and initialized the WRF-Chem SCM model with averaged BC profile taken from three-dimensional simulation results for the same case by Ding et al. (2016), which had been verified to perform well in capturing the temporal variations of BC. The inputted BC vertical profile in Fig. 1 indicated that during that case, BC concentration reached up to approximate 30 μg m$^{-3}$ near the ground surface and decreased rapidly along altitude to a relatively constant value of less than 5 μg m$^{-3}$ above 800 m. However, although BC profile featured the maximum concentration near surface, the heating efficiency of BC due to shortwave radiation absorption peaked around 600-800 m, indicating that BC in the upper PBL is more efficient in terms of absorbing shortwave



radiation and heating surrounding air masses. Suck kind of heating profile is partly caused by the fact that incident solar radiation is attenuated by aerosols, trace gases and cloud when transferring in the atmosphere. Therefore, BC at higher altitude tended to be subjected to higher incident radiation flux and absorb more solar energy. Meanwhile, lower air density made the upper air more readily to be heated compared with that near the surface. Due to light absorption caused by BC together with

additional extinction caused by scattering aerosols, solar radiation reaching the surface was diminished to a relatively large extent, resulting in less sensible heat flux thus lower temperature in near-surface atmosphere.

Overall, upper-level warming and surface cooling substantially modified the temperature stratification. As illustrated in Fig. 2, the afternoon upper-level heating and morning surface cooling could be clearly identified. The strongest upper-air warming had sustained to be over 1.0 °C between 1000 m -1200 m since mid-noon (12:00LT) because the incident shortwave radiation

was most intensive and heating effect had already been accumulated through forenoon. The strongest cooling effect appeared in the morning (9:00-11:00 LT) and decreased very sharply after noontime. The reason is that, with increasingly intense turbulence during morning boundary evolution, part of the surface cooling was gradually compensated by the entrainment of more warmed air into the turbulent PBL from above when the PBL was developed in the late morning. The warming and cooling of different atmospheric levels remarkably altered the stratification, thereby weakening convective motions. Stable

stratification combined with decreased sensible heat flux at the ground surface greatly suppressed vertical turbulence in the boundary layer (Wilcox et al., 2016), contributing to a delay of PBL development and an earlier drop as well as a substantial decrease in PBL height, which hindered the air pollutants from being further dispersed vertically.

**3.2 Impacts of altitude and concentration of BC aerosol layer**

As aforementioned, specific synoptic condition, chimney plumes and regional transport of air pollution could result in non-

uniformly distributed pollutant profiles (Xu et al., 2014;Guinot et al., 2006;Ding et al., 2009). Vertically inhomogeneous distribution of BC aerosol has been frequently observed by in-situ aircraft and tethered balloon measurements (Zhao et al., 2015;Li et al., 2015). Given that BC was measured to rise around 800-1000 m during daytime of haze episode (Li et al., 2015) and the upper-PBL BC has higher light-absorbing efficiency and solar heating usually maximizes, BC plume with the maximum concentration of 10 μg m$^{-3}$ at central axis is taken as a typical example to illustrate the perturbations on detailed

physical processes related to PBL evolution due to vertically non-uniform BC profile. As displayed in Fig. 3a, with the existence of absorptive BC, incident solar radiation at the surface is diminished by about 5.9 W m$^{-2}$ at 12:00 LT, leading to a decline in surface temperature of 0.2 °C thus surface sensible heat flux of 3.6 W m$^{-2}$, which would otherwise heat lower atmosphere, promote turbulent motions and enhance the PBL development. Meanwhile, absorbed radiation energy is converted to thermal energy and leads to a substantial heating near the top of PBL, as reflected by a rise in air temperature around 1.0 °C.

Accordingly, modified temperature profile results in a more stable stratification with its exchange coefficient falling by 15% (Fig. 3b), i.e. less turbulence mixing. Similarly, a substantial decrease of PBL height is attributed to less surface heat flux accompanied by stable stratification, which is showed in Fig. 3c.



To shed more light on impacts of various BC vertical distribution on the dome effect, we conducted hundreds of parallel numerical experiments by changing the altitude and concentration magnitude of BC from the simulation shown in Fig. 3 to a larger extent to figure out how the PBL evolution responses to varying BC profiles and mass loading. Specifically, we manually increase BC concentration from 0 to 30 μg m$^{-3}$ with an increment of 5 μg m$^{-3}$, all of which are placed at the altitude from 150

m to 2250 m with a stride of 300 m. Thus, the multidimensional experiment consists of 105 parallel numerical simulations. All the individual simulations (marked by black dots in Fig. 4) and resultant perturbations on PBL height and turbulence coefficient under different altitude and concentration loading of BC are shown in Fig. 4. As several studies have pointed out, BC near the ground surface warms the earth-atmosphere system and favors the development of PBL by trapping more solar radiation that supposed to be reflected by land and then promoting the convective motions (Huang et al., 2015;Barbaro et al.,

2014). The increased air temperature near surface weakens the capping inversion, acting to cancel the effect of reduced buoyancy flux at ground surface and raising the top of PBL (Yu et al., 2002). However, this enhancement in PBL development is not that noticeable in magnitude while compared with PBL suppression due to BC at higher altitude. Consistent with several existing observational and numerical studies, absorbing smoke aloft is capable of remarkably changing the energy balance between the surface and the atmosphere in a way that stabilizes the boundary layer and suppresses convection (Koren et al.,

2004;Ackerman et al., 2000). As demonstrated in Fig. 4, the PBL top could be decreased by about 15% and the turbulent exchange coefficient dropped by over 20% due to high-altitude BC plume. The substantial suppression effect induced by the BC plume maximizes around the top of PBL can be attributed to two main reasons. Firstly, incident solar radiation flux at the top of PBL is usually most intensive. After being absorbed by BC, it can effectively heat the surrounding air and change the strength of capping inversion. Secondly, turbulent exchange coefficient falls to a relatively small value while getting close to

the top of PBL (Fig. 3b), indicating that vertical turbulence exchange for heat at this height is rather weak. Therefore, the warming layer is prone to being kept at that level rather than diffusing vertically, further strengthening the capping inversion and consequently lowering the PBL height.

Another interesting fact is that, while BC layer locates above the original top of PBL (i.e. above the 1300 m in Fig. 4), the boundary layer also becomes shallower but relatively less notable compared with upper-PBL BC layer (i.e. 600-1000 m). In

this case, BC also blocks part of the incoming solar radiation and diminishes surface fluxes proportionally, nevertheless, it no longer alters the temperature stratification of PBL below. As commonly known, changes in the height of PBL are determined by the surface buoyancy flux and the capping inversion, and both of them are affected by BC plume (Ding et al., 2017;Ding et al., 2016). Since the impact of column loading of BC to surface buoyancy flux will not change a lot in the lower troposphere, here the different impact of BC on PBL height around the capping inversion (i.e. 600-1200m) with the above altitudes indicates

that the upper-air warming due to BC layer plays a more dominant role in depressing the PBL height and hence enhancing the near-surface air pollution.



### 3.3    Amplified dome effect by mixing with scattering aerosols

During wintertime heavily polluted episodes while both primary and secondary pollutants increases dramatically, BC will be coated by scattering aerosols through condensation of low-volatility gases and coagulation with secondary aerosols, and thus becomes hydrophilic and more internally mixed (Shiraiwa et al., 2007;Chen et al., 2016). Theoretically, its absorption

properties can be significantly enhanced by internal mixing with other compounds because the coatings act as a lens and enlarge its mass absorption cross section effectively (Bond et al., 2013;Chen et al., 2017a). We analyzed this absorption amplification effect based on the consistent in-situ measurements during the development of severe haze pollution in Beijing in January 2013 (Sun et al., 2014). Compared with clean period, the secondary aerosol increased significantly during haze episodes. Take sulfate aerosol for instance, its concentrations during hazy days were approximately 55 times higher than those

during clean condition. Based on the ratio of BC to SNA (sulfate, nitrate and ammonium) during different periods and assuming BC concentration is 5 $\mu g \, m^{-3}$ (Zhang et al., 2013), absorption at the simulated wavelength (i.e. 300, 400, 600 and 1000 nm) was amplified by factors of 1.8, 1.7, 1.6 and 1.4, respectively (Fig. 5). These absorption coefficient amplification factors are comparable with previous laboratory and in-situ observational studies, which suggests a range of absorption enhancements from 1.05 to 3.50 due to BC mixing morphology (Bond et al., 2013;Shiraiwa et al., 2010;Schnaiter, 2005;Peng et al., 2016).

Accordingly, shortwave heating rate also increased from about 0.15 $K \, h^{-1}$ to over 0.22 $K \, h^{-1}$, which considerably accelerates the warming effect induced by BC (Fig. 5).

In addition to directly increasing absorption cross section, SNA could undergo deliquescence and lead to an increase in aerosol diameter to a large extent, especially under humid conditions, indirectly enhancing the light-absorbing capacity (Tsai and Kuo, 2005;Zheng et al., 2015;Liu et al., 2011). To comprehensively understand this enhancement effect induced by SNA, we

conducted tens of sensitivity experiments under different level of SNA concentration and relative humidity (RH). As presented in Fig. 6, aerosol absorption extinction coefficient increases with SNA concentration. This dependence on SNA is much notable under lower RH (50%) before aerosol starts to deliquescence. By contrast, when the air gets more humid, aerosol water uptake is getting increasingly important. According to round-year observational data (Zhang et al., 2012), SNA concentrations and humidity condition varies a lot across in China. For typical cities in China, the annual averaged SNA level ranges from 40-60

$\mu g \, m^{-3}$ in Nanjing (118.95°E, 32.12°N) and Shanghai (121.45°E, 31.22°N) to almost 140 $\mu g \, m^{-3}$ in Beijing (116.30°E, 39.99°N), but the annual mean BC concentrations are approximately 5 $\mu g \, m^{-3}$(Ye et al., 2003;Zhang et al., 2013). In Nanjing with lower SNA level but higher RH, SNA induced hygroscopic growth may play the dominant role in the enhancement of aerosol light-absorbing efficiency. Instead, relatively higher SNA concentration in Beijing makes the main contribution according to Fig. 6. Specifically, the absorption could be elevated by 63% when RH rises from 50% to 90% in Nanjing, while for Beijing, the

corresponding enhancement is only 18%. Such disparities indicate that aerosol absorption and further impact on boundary layer evolution have a closer link to humidity and will be more important in coastal region at lower latitudes in China.

Enhanced light absorption and heating efficiency by mixing with SNA certainly perturbs aerosols-PBL interactions (Fig. 7). Overall, the PBL responses tend to be magnified in terms of both increased PBL top by near-surface BC and suppressed





convections by upper-level BC. For typical wintertime meteorological conditions in Beijing, aging processes could lead to a decline of 8 W m$^{-2}$ in surface heat flux and additional 25% decrease in PBL top in maximum when compared with those caused by freshly emitted BC. What is more, when BC concentration exceeds 20 μg m$^{-3}$, this absorption enhancement lead to the phenomenon that even BC at lower altitude may increase the stratification and exert negative effect on PBL development (See

the right-hand part of Fig. 7). In other words, there possibly exists a critical point of BC concentration for its associated dome effect, same as the non-linear relationship of dome effect to BC emission in different megacities presented by Ding et al. (2016). BC mitigation under this point would largely weaken the impact on PBL suppression attributed to smaller BC loadings and less influence by mixing with scattering aerosols. For the perspective of the policy making on air pollution mitigation, top priority ought to be given to BC emission reductions, especially during heavily polluted day when the relative humidity was

increased associated with the cooling within the PBL due to the aerosol-PBL feedback (Ding et al., 2017;Ding et al., 2013;Ding et al., 2016).

### 3.4 A comparison of the dome effect over urban and rural areas

Heretofore, BC induced aerosol-PBL interaction has been investigated mostly for cites. However, urban area only takes up approximately 5% area in China while cropland is the dominant surface type also featuring dense population and high PM$_{2.5}$

exposure (Liu, 2005;Xu et al., 2011). Frequent regional-scale haze pollution and cropland-dominant land cover in China make it necessary to figure out this interaction over rural area (Yang et al., 2015;Zhao et al., 2013b). The differences in surface parameters between cities and rural area are listed in Table 2. Comparatively, rural surface, including cropland and pasture, is characterized by larger surface albedo, higher soil moisture and greater heat capacity.

Radiation energy balance without any influence from aerosol already shows great disparities for the two kinds of surface land.

Under the same level of incident solar radiation intensity, net shortwave radiation at ground surface is mainly determined by land surface albedo. In comparisons with urban area, less net downward solar radiation on rural surface is ascribed to larger albedo reflecting more solar radiation, accompanied by higher upwards shortwave radiation flux at the top of the atmosphere, as shown in Fig. 8. Correspondingly, the surface temperature in rural area at noontime is almost 1.5 °C colder than that in urban area, indicating a strong urban heat island effect (Oke, 1982). As for the atmospheric heating, there is little difference

over both surface types when excluding the radiative perturbation of aerosols. The resultant height of PBL is expected to be 250 m lower in rural area.

If taking aerosol radiative effect into account when BC is 10 μg m$^{-3}$ around the altitude of 1000 m, over both urban and rural area, net downward shortwave radiation flux, surface temperature and upwelling shortwave radiation show a notable decrease. At the same time, atmospheric heating rate in the upper air increases in response. The most distinctive difference between

these two kinds of land surface is that upwelling shortwave radiation flux at the top of the atmosphere over rural surface shows a reduction twice as much as over urban surface, indicating a larger portion of solar energy is blocked in the atmosphere, which corresponds to much stronger atmospheric heating rate due to radiation absorption. Meanwhile, it suggests a slightly larger decrease in surface temperature over rural surface, followed by a less sensible heat transfer into atmosphere. The greater surface



cooling together with more intensive upper warming jointly enhance the stable stratification. Therefore, the decrease in PBL top in rural area doubles that in cities, as shown in Fig. 8.

The comprehensive analysis on responses of PBL to various magnitude and altitude of BC aerosol over urban and rural land surface is illustrated in Fig. 9. As mentioned before, since PBL is less-well developed in rural area, the altitude of BC plume

with maximum suppression effect on PBL was a bit lower than that over cities. Additionally, BC induced heating is more intensive over rural surface due to the fact that larger albedo results in more shortwave radiation reflected by the ground surface, part of which was absorbed by BC aerosol when transferring upwards. Consequently, the promotion and suppression effect on PBL attributed to BC aerosol over rural surface is about 15% larger than that over urban surface. That is, BC would exert a more intensive dome effect over rural surface. It should be noted that residential combustion of raw coal and biofuel in a small

domestic stove in rural area is responsible for the majority of BC emission in China (Zhi et al., 2008;Li et al., 2017a). More stable PBL and more significant dome effect over rural surface would further favor the formation and accumulation of regional air pollution.

## 4. Summary and implications

The dome effect of BC plays a crucial role in air pollution deterioration, which is expected to be highly dependent on many

factors like vertical distribution, mass loading, and aging processes of BC as well as the underlying land surface. By integrating available in-situ observations of meteorology and atmospheric aerosols together with simulations using meteorology-chemistry online coupled model WRF-Chem SCM, we conduct a set of multidimensional experiments, each of which contains hundreds of parallel simulations, to quantify the impacts of these factors in enhancing the surface haze pollution. We found that the dome effect of BC is extremely sensitive to the altitude of aerosol layer. In more detail, near-surface BC tends to promote the

PBL development while BC aloft is more essential in the PBL suppression, especially those near the capping inversion. Meanwhile, this work indicates that the dome effect of BC can be significantly intensified when BC gets internally mixed with scattering aerosol during winter haze event, which could further decrease PBL height by up to 25%. In terms of different underlying surface, the dome effect is more substantial in rural areas. Under the same condition of BC, PBL top decrease in rural area could be 15% greater than the corresponding value in the urban area.

Haze pollution in China is getting increasingly frequent and thus has received extensive attentions from the public and government. In spite of a series of emission control measures, the air quality has never been essentially improved so far, especially for the extreme haze events in winter. Our study highlights the importance of BC in worsening air pollution and further provides clues for both long-term policy making on air quality improvement and short-term pollution emergency plan. Since that BC from the elevated emission sources, i.e. point sources with high chimneys like coal-fired power plants and

industrial boilers would deteriorate dispersion conditions by suppressing PBL, it is an efficient way to mitigate near-surface air pollution to preferentially reduce BC emission from these sources during severe haze pollution. In the longer run, clean energy substitution and extending of BC emission reduction technologies should be encouraged. In addition, it is noteworthy



that residential combustion in rural area is responsible for the majority of BC emission in China, such significant dome effect over rural surface would further favor the formation and deterioration of regional air pollution. As a result, it can be implied that abatement technology for domestic stove in rural area could be another cost-effective way to reduce regional air pollution. Of course, as one of the key short-lived climate forcers, BC has been found to be one of the most important components

contributing to the global warming, these efforts devoted in reducing BC emission in both urban and rural areas in China definitely result in a substantial reduction of the national carbon emission and co-benefit the mitigation of global warming.

In addition, our study here highlights the significance of investigating the vertical structure of the detailed atmosphere processes in the lower atmosphere, which is very important for improving the understanding of interaction of atmospheric physics and chemistry. However, in China, most of the existing and current efforts of field measurements, including both

routinely operated monitoring networks and research based field measurement stations, have been mainly focused on the ground surface. There is very limited information of the vertical profiles of atmospheric aerosols and physical parameters related to key processes in the PBL. For numerical models, there is also a lack of vertical dataset for the model evaluation and parameterization improvement. To gain a more comprehensive understanding of the causes and evolution of air pollution, more vertical measurements, including in-situ measurement based on aircraft and tagged balloon platforms etc., and remote

sensing measurements using ground-based instruments and satellites are urgently needed in the future.

**Acknowledgements**

This work was supported by the National Natural Science Foundation of China (D0512/91544231, D0510/41725020, and D0510/41505109). Part of this work was supported by Ministry of Science and Technology of the People's Republic of China

(2016YFC0200506), the Public Welfare Projects for Environmental Protection (201509004) and Jiangsu Provincial 2011 program (Collaborative Innovation Center of Climate Change).



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




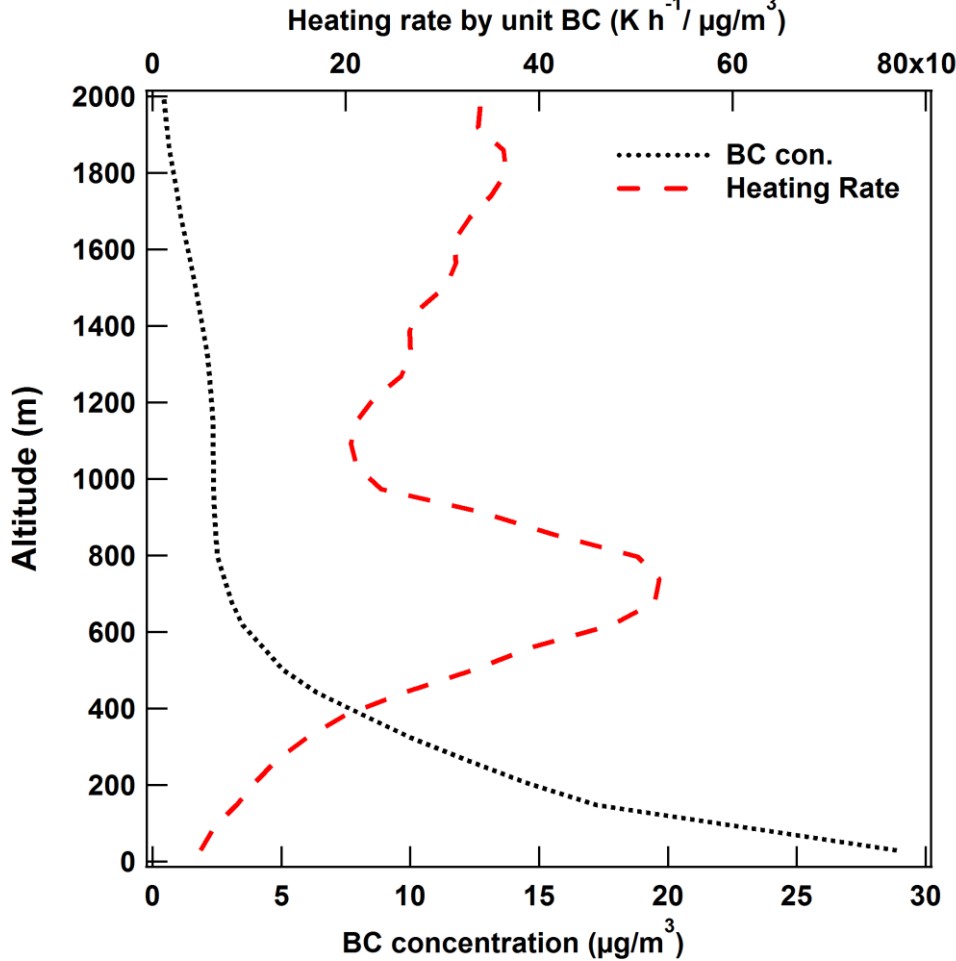

**Figure 1. Vertical profile of BC and shortwave heating rate included by unit BC mass in the afternoon (12:00-16:00 LT) during a heavy polluted episode 23rd-24th December, 2013 in Beijing. BC profile was extracted from regional WRF-Chem modeling by Ding et al. (2016).**

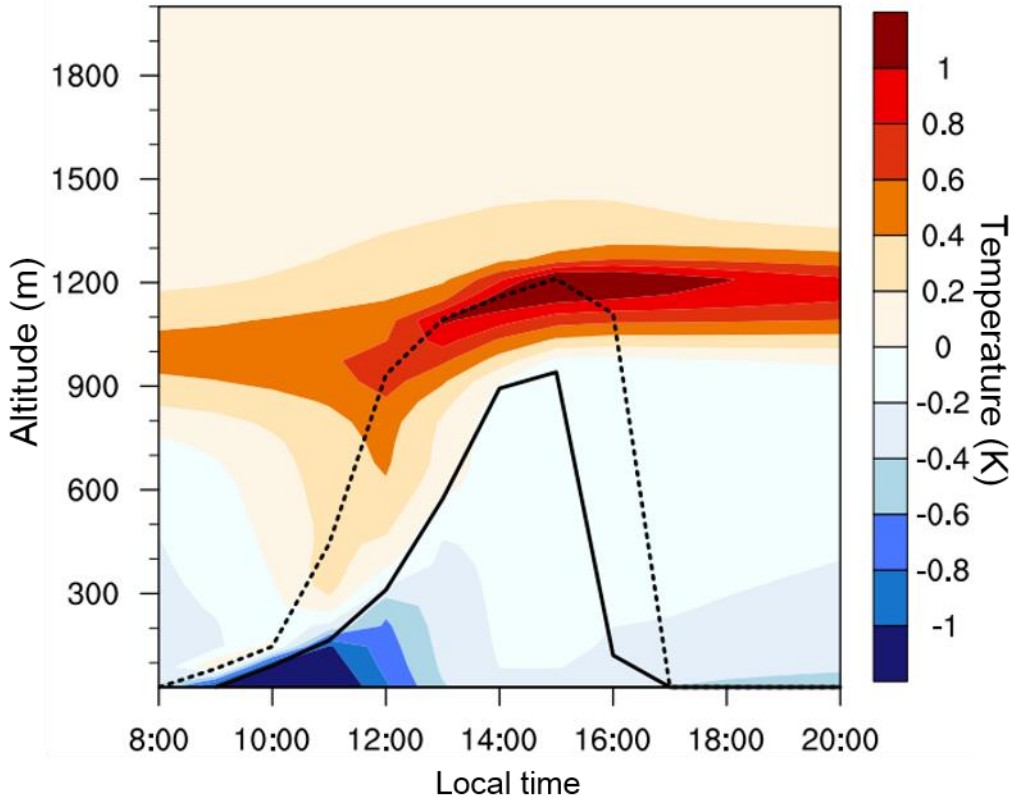

**Figure 2.** Diurnal variations of the air temperature change caused by aerosols during haze episode in Beijing and of PBL height for runs with (solid line) and without (dash line) ARI.



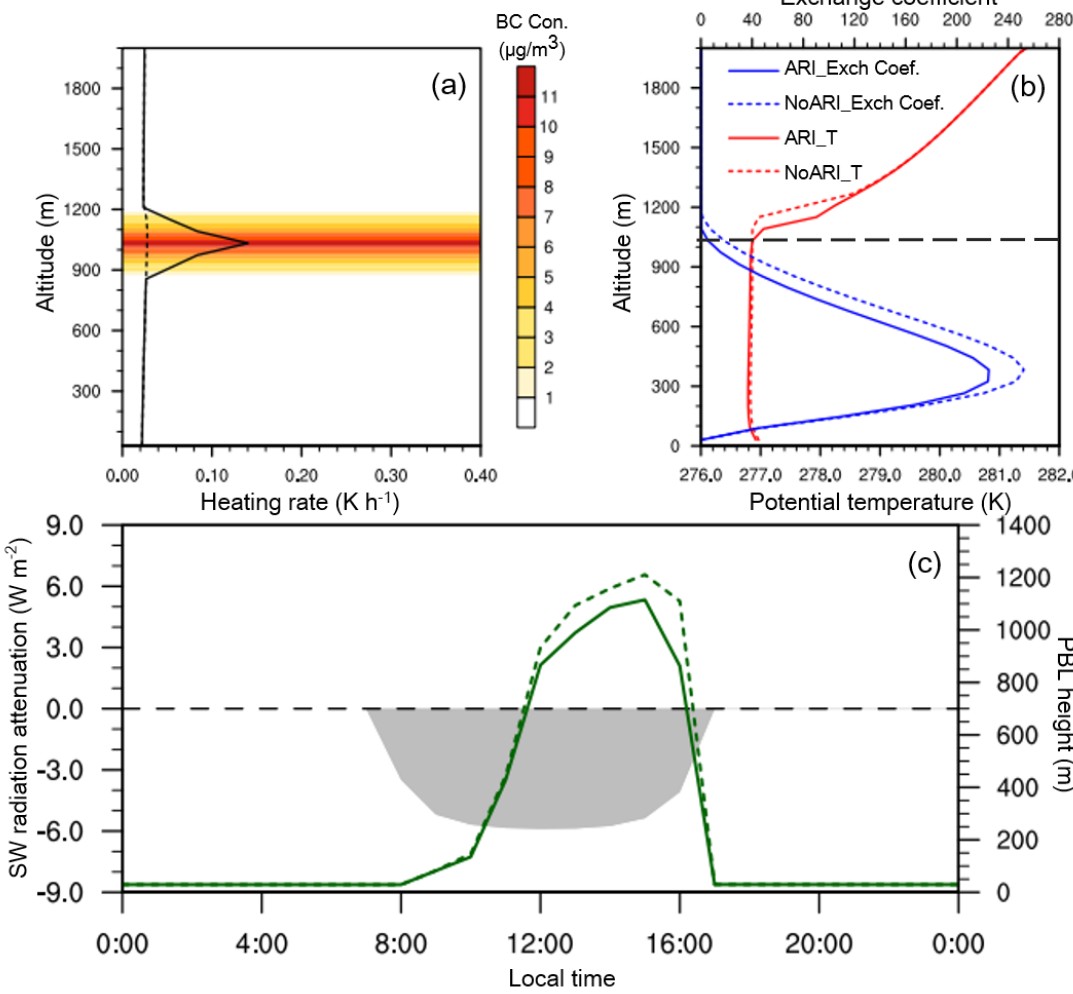

**Figure 3.** (a) Vertical distribution of BC aerosols (contour) and shortwave heating rate at 14:00LT for runs with (solid line) and without (dash line) ARI. (b) Vertical potential temperature profile (red) and exchange coefficient profile (blue) for runs with (solid line) and without (dash line) ARI. (c) Diurnal variations of PBL height for runs with (solid line) and without (dash line) ARI and shortwave radiation attenuation at surface induced by BC absorption.



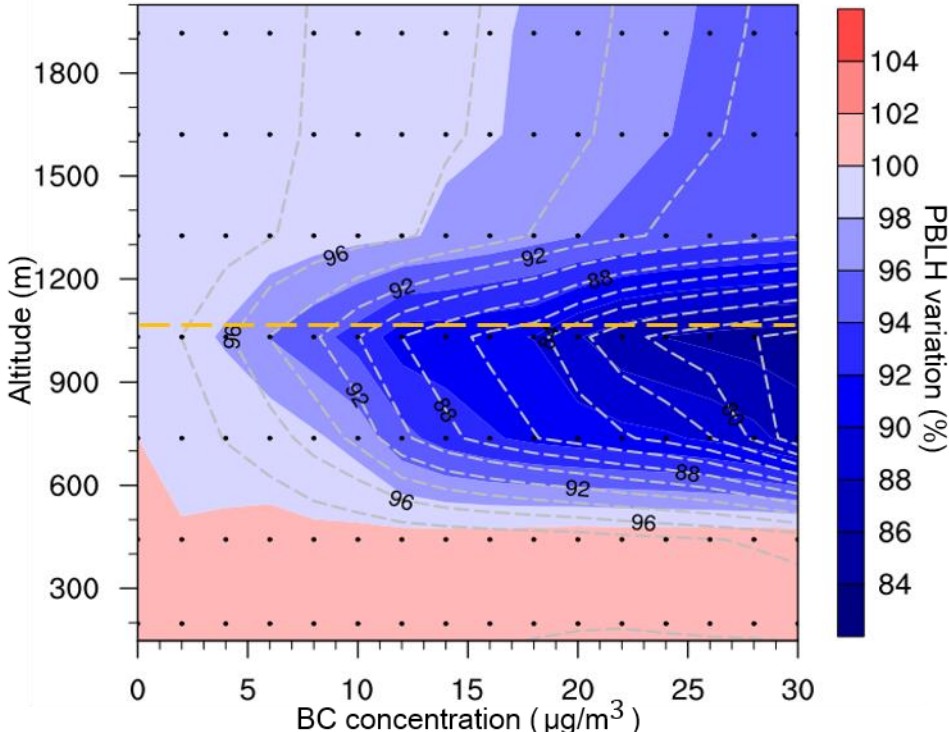

**Figure 4. PBL height (contour map) and turbulence exchange coefficient (grey dash isoline) variations in percentage as a function of the altitude and mass concentration of BC aerosol layers. Note: each dot on the figure represents an experiment with different BC input. X-axis gives concentrations in the center of the inputted BC plume, Y-axis gives the altitude of the residing plume. Yellow dash line represents the original PBL height without any BC plume.**





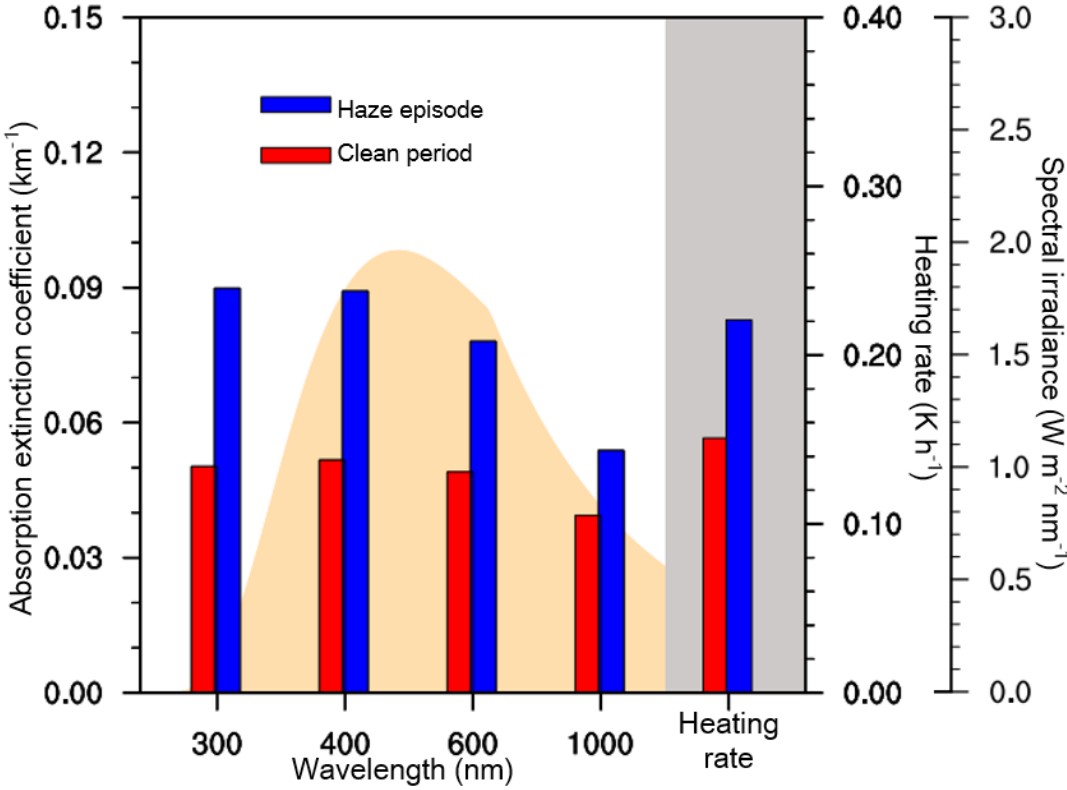

**Figure 5.** Aerosol absorption coefficient at different wave length (300,400,600, and 1000nm) and shortwave heating rate at 12:00LT for runs during clean period and haze episode when the ratio of BC and SNA is 1:3 and 1:8, respectively. The bright yellow shadow schematically gives solar spectral irradiance at sea level.



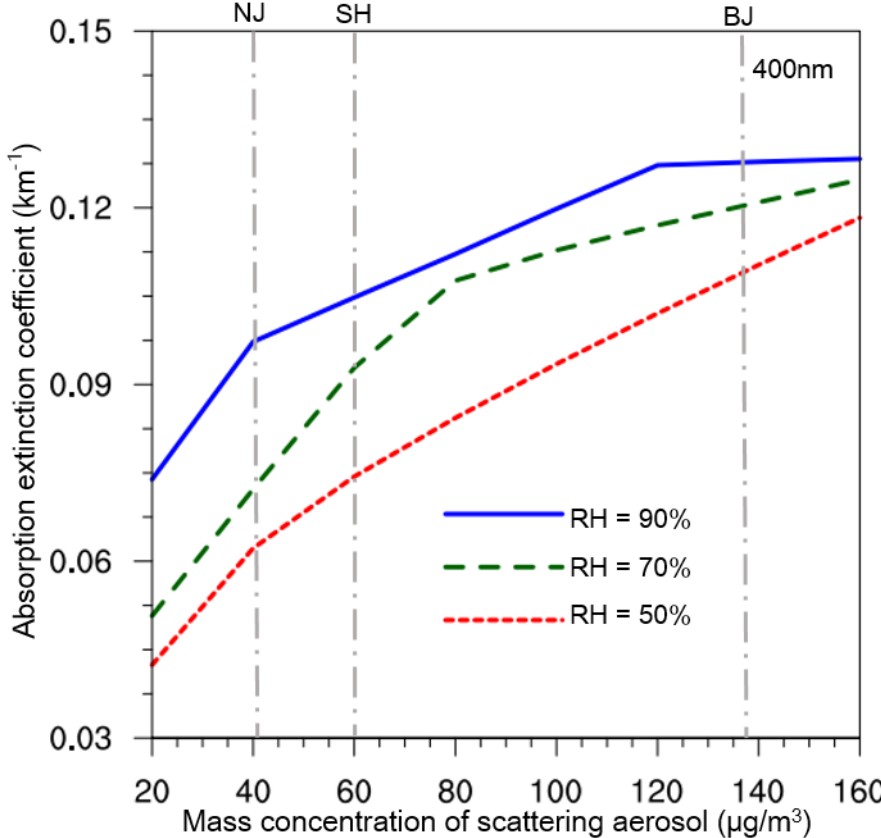

**Figure 6. Variation of absorption extinction coefficient as a function of different mass concentrations of scattering aerosols mixing with BC. The BC concentration is fixed at 5 µg/m³ under various moisture conditions of relative humidity being 50%, 70%, 90%. Three main cities (BJ, SH and NJ) is marked in the figure with their annual mean PM$_{2.5}$ concentrations.**





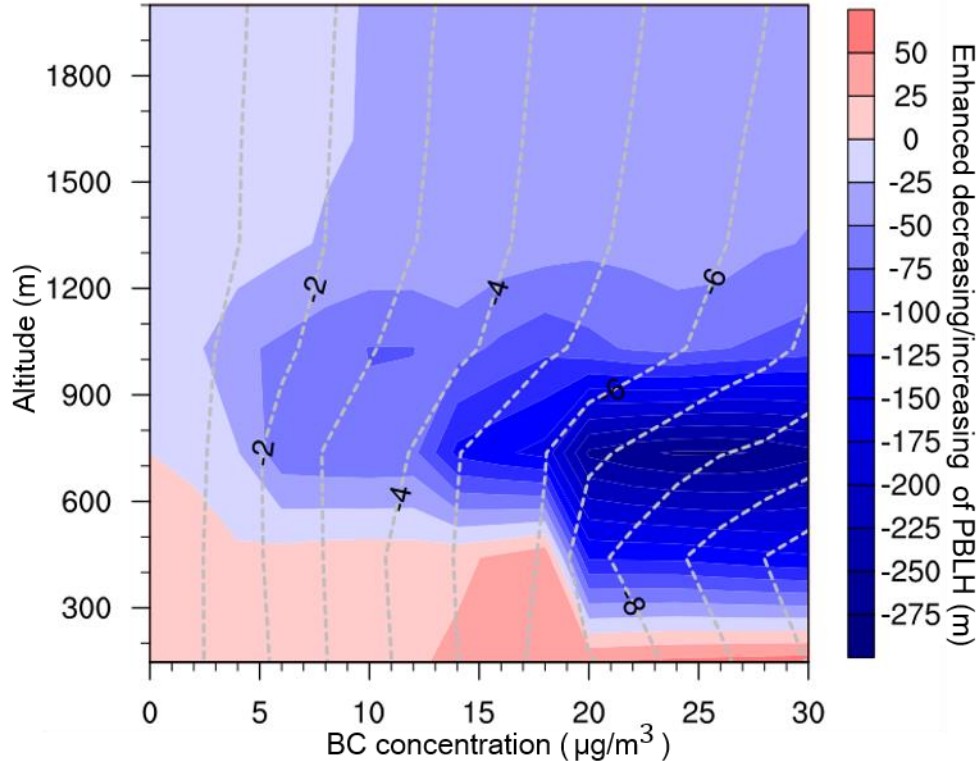

**Figure 7.** Enhanced decreasing/increasing of PBL height (contour map) and enhanced reduction of sensible heat flux at surface (dash isoline) caused by amplified absorption of BC internally mixed with scattering aerosols. The mass concentrations of BC, $SO_4^{2-}$, $NO_3^-$, $NH_4^+$ possess a ratio of 1:3:2:1, which is extracted from observational concentrations for Beijing annual $PM_{2.5}$ and selected species. X and Y axes are the same as Fig. 4.



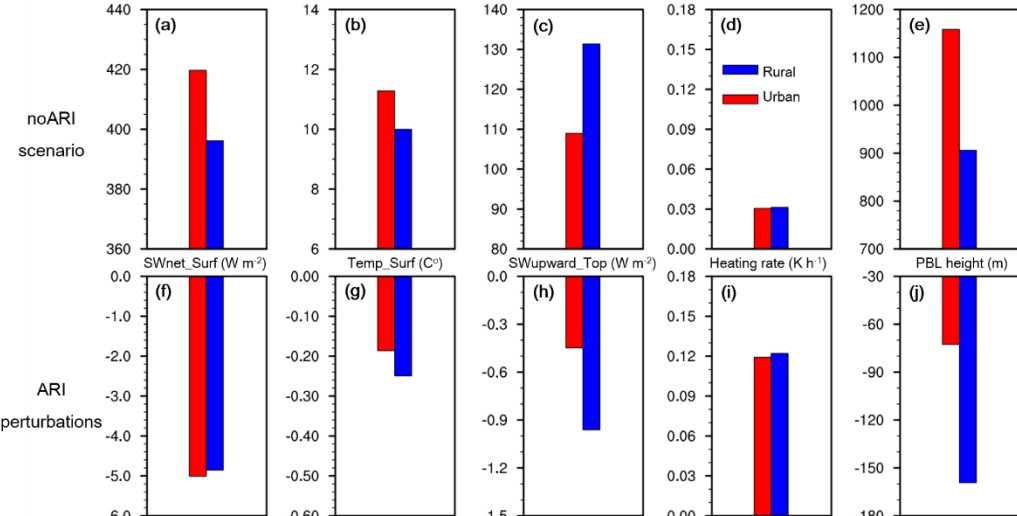

**Figure 8. Net downward shortwave radiation at surface (SWnet_Surf), surface temperature (Temp_Surf), upwelling shortwave radiation at model top (SWupward_Top) and shortwave heating rate, PBL height for urban and rural underlying surface without aerosol effect (a, b, c, d, e), and their corresponding perturbations due to aerosol-boundary layer interaction (f, g, h, i, j).**




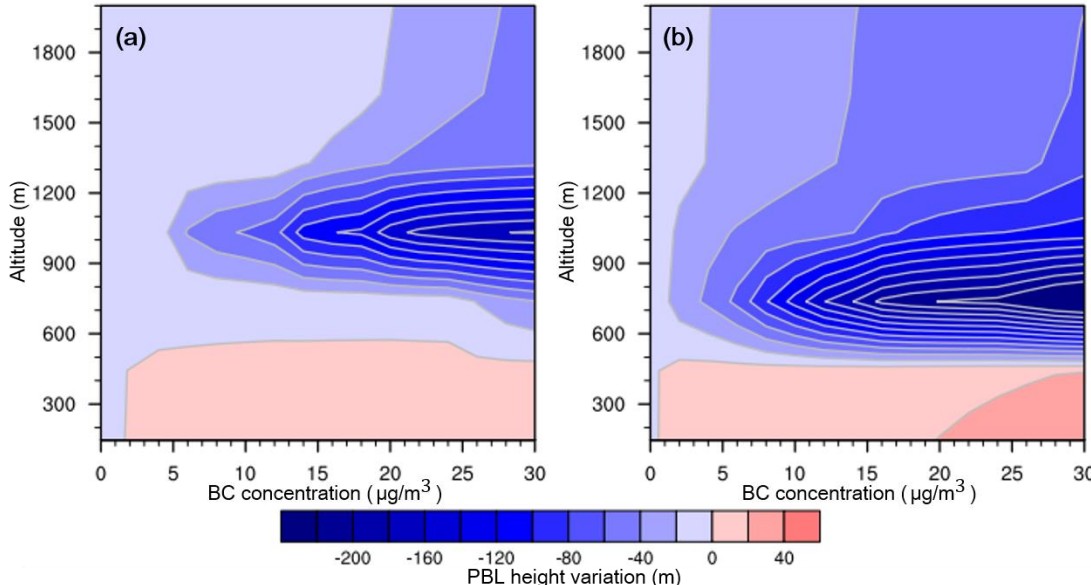

**Figure 9. Different PBL height variations due to specific BC distribution over (a) urban surface and (b) rural surface. X and Y axes are the same as Fig. 4.**



**Table 1. WRF-Chem domain settings and configuration options**

| Domain setting | |
|---|---|
| Horizontal grid | 3×3 |
| Grid spacing | 4 km |
| Vertical layers | 100 |
| Configuration options | |
| Longwave radiation | RRTMG |
| Shortwave radiation | RRTMG |
| Cumulus parameterization | Grell-Deveny |
| Land surface | unified Noah |
| Boundary layer | YSU |
| Microphysics | Lin et al. |
| Gas-phase chemistry & aerosol scheme | CBMZ and MOSAIC |



**Table 2. Distinctive differences of urban and rural land cover**

| Properties | Urban and built–up land | Irrigated cropland and pasture |
| --- | --- | --- |
| albedo | 0.15 | 0.2 |
| Soil moisture | 0.1 | 0.5 |
| emissivity | 0.88 | 0.985 |
| roughness | 80 | 2 |
| thermal inertia | 3 | 4 |
| heat capacity | 18.9e5 | 25.0e5 |