# Peer review of "Dome effect of black carbon and its key influencing factors: A onedimensional modelling study"

_Atmospheric Chemistry and Physics, 2017_

## Referee Comment (RC1) · Anonymous Referee #1 · 13 Nov 2017

The authors used WRF-Chem single-column model combining with available in-situ observation to quantify the impacts of some key factors on BC's dome effect. They demonstrated that the dome effect of BC aerosols strongly depends on the vertical distribution and aging process of BC, as well as the underlying land surface. The technical work appears to be competent. The paper contains important addition to existing literature, and is suited for publication to ACP. However, there are several concerns that should be addressed or considered before being accepted for publication.

(1) Model configuration: In terms of mixing state of multiple aerosol chemical compositions, the authors presume that the aerosols in each bin were internally mixed, which

would overestimate the light-absorbing capability of BC aerosols. In the atmosphere, a large amount of aerosol mass was partitioned onto BC-free particles rather than internally mixing with BC.

(2) P5/L17: When the abbreviation regarding "SNA" first appeared in the paper, the author should also give its full terms.

(3) Why the author only considered BC mixing with SNA? Why not consider other aerosol compositions, especially organic species that play an important role in the mass fraction of aerosol particles.

(4) Figure 1 shows the heating efficiency of BC due to shortwave radiation absorption peaked around 600-800 m. But the strongest upper-air warming exhibited at 1000-1200 m shown in Fig. 2, different with the height of the largest heating efficiency of BC. The author need to explain the reason.

(5) According to the vertical profile of shortwave heating rate included by unit BC mass (Fig. 1), the authors discussed that BC in the upper PBL is more efficient in terms of absorbing shortwave radiation and heating surrounding air masses. To support the author's statement on the upper PBL, Figure 1 need to show the average PBL height in the afternoon (12:00-16:00 LT) during a heavy polluted episode 23rd-24th December, 2013 in Beijing.

(6) P6/L25-27: The authors state here that Figure 3a displayed the information of the incident solar radiation at the surface, decline in surface temperature and surface sensible heat flux. I did not found these information in Fig. 3a. The authors should check.

(7) In Fig. 3b, what does the black dash line represent? Authors should state it in the caption of the figure 3.

(8) P7/L4: According to the data shown in Fig.4, the increment of increase BC concentration from 0 to 30 $\mu$g m-3 should be 5$\mu$g m-3 rather than 2 $\mu$g m-3.

(9) P8/L10-15: The calculation method of absorption coefficient amplification factors of

BC was different with that in the literature. Authors calculated the absorption amplification of BC aerosols based on the BC/SNA ratio during different periods (i.e., clean period: BC/SNA=1:3, haze episode: BC/SNA=1:8). The obtained absorption amplification (i.e, 1.8, 1.7, 1.6 and 1.4 at wavelength 300, 400, 600 and 1000 nm) was then compared with previous laboratory and in-situ observational studies. However, the absorption amplification of BC aerosols in previous studies represents that the enhancement of light absorption of BC-containing particles due to coating materials on BC surface comparing with bare BC.

(10) In Fig.5, authors used BC/SNA ratio of 1:3 and 1:8 to calculate the aerosol absorption coefficient at clean period and haze episode, respectively. They should give the data sources of BC/SNA ratio at different periods. Moreover, Fig.5 also shows the solar spectral irradiance at sea level (bright yellow shadow). However, there was not any discussion on this information in the paper. Why did the authors give solar spectral irradiance at sea level in Fig. 5.

(11) Considering most of days in north China characteristic of RH lower than 50%, I suggest adding a case of RH=30% in Fig.6. Moreover, authors gave the absorption extinction coefficient at a certain wavelength (i.e., 400 nm), they should state it in the caption of the figure 6.

(12) RH at a certain SNA influence not only aerosol particle diameter but also their reflective index (RI). Authors mentioned the change of aerosol particle diameter at different levels of RH (P8/L17-18). Did they also consider the RI change? The authors should give detail description on data process at different levels of RH in Sect. 2 (data and method), such as the diameter growth factors and RI setting in the model calculation.

(13) When investigating the impacts of BC aging process on light absorption, authors discussed the difference of absorption under difference BC/SNA ratio, shown in Fig.5 and Fig.6. However, why they fixed a value of BC/SNA ratio (i.e., 1:6 shown in Fig.

7) rather than using different BC/SNA ratio as above two figures to further discuss the impacts of BC aging process on PBL height? Figure 5 shows different BC/SNA ratio under different pollution levels (i.e., clean period: BC/SNA=1:3, haze episode: BC/SNA=1:8) based on in-situ observation from literature, indicating that BC/SNA ratio will change with BC concentrations. Therefore, it seems to be unreasonable to assume same BC/SNA ratio under different BC concentrations.

(14) When BC concentration is lower than 2 $\mu$g m-3, the PBL height variations due to specific BC distribution shown in Fig. 9 exhibited significant difference with that shown in Fig.4. Why?

---

## Referee Comment (RC2) · Anonymous Referee #2 · 15 Nov 2017

This study investigated the black carbon "dome effect" and its key influencing factors, namely the vertical distribution and aging processes of BC, and the underlying land surface. The "dome effect" can play an important role in haze evolutions, which makes this study an interesting topic. Also, the manuscript is well organized and clearly presented, and is worth publishing. However, several concerns need to be addressed before the final publication.

Major comment:

(1) One major concern of this study is the lacking of information on actual scenarios. While low-level BC can enhance PBL height while upper-level would suppress that,

what is the approximate threshold of low-level BC to upper-level BC concentration ratios, at which these two effects can offset each other? Is this threshold easily reached during haze events? That is, how often and how universal is the "dome effect" present? In actual scenarios, the BC are more likely to be composed of both a low-level freshly emitted peak, and an upper-level transported peak. Their different ratios may lead to different overall effect. Since observation on vertical BC profile is scarce, a relatively long-term simulation covering a larger domain (e.g., northern and eastern China) with actual configurations like the one shown in Fig. 1 might be helpful, or at least this issue should be discussed in more detail.

(2) Although the simulation results are well explained, the conclusion about chimneys and domestic stoves seems somewhat abrupt. What is the typical height of chimneys? Can that compare to the height of the inversion layer? It was more confusing on the conclusions about domestic stoves at rural areas. In the context of this manuscript, the depression of PBL at rural areas should be caused mainly by the long-range transported upper-level BC, not the local emitted ones. On the contrary, the freshly emitted BC would serve as the low-level BC and tend to enhance the PBL. Thus the fact that rural areas are more sensitive to "dome effect" would lead to the conclusion that reducing long-range transported upper-level BC is more important. The casual relationship should be better described.

Minor comments:

Page 2 Line 1: "developed regions like...": change into "the more developed regions like..."

Page 2 Line 6: is the "680 ug/m3" daily average? Later the hourly maximum of $\sim$900 ug/m3 is mentioned, so here need some clarification.

Page 2 Line 17-L18: consider change the expression of "concentration of BC... far more than..."; "more concentration" seems strange.

---

## Author Comment (AC1) · 9 Jan 2018

**Response to Referee #1**

*General Comments: The authors used WRF-Chem single-column model combining with available in-situ observation to quantify the impacts of some key factors on BC's dome effect. They demonstrated that the dome effect of BC aerosols strongly depends on the vertical distribution and aging process of BC, as well as the underlying land surface. The technical work appears to be competent. The paper contains important addition to existing literature, and is suited for publication to ACP. However, there are several concerns that should be addressed or considered before being accepted for publication.*

**Response:** We would like to appreciate the referee for providing the suggestions. We have revised this article according to the comments.

*Specific Comments: 1. Model configuration: In terms of mixing state of multiple aerosol chemical compositions, the authors presume that the aerosols in each bin were internally mixed, which would overestimate the light-absorbing capability of BC aerosols. In the atmosphere, a large amount of aerosol mass was partitioned onto BC-free particles rather than internally mixing with BC.*

**Response:** Thanks for raising this import point. Yes, in this work aerosol components in each size bin are assumed to be internally mixed, which means that BC is uniformly distributed throughout the particle and its light-absorbing efficiency could be amplified by other scattering aerosols. Many existing in-situ measurements suggested that the majority of carbonaceous particles were internally mixed, especially during hazy days, and were highly influenced by secondary species in eastern China (Li et al., 2011;Wang et al., 2014b;Zhang et al., 2013). Single-particle soot photometer measurements revealed that the fraction of internally mixed BC was variable and could be as high as around 70% in the Pearl River Delta region and approximately 50% in Beijing area (Wang et al., 2014b;Huang et al., 2012). Accordingly, BC absorption enhancement with nitrate and sulfate is also frequently observed in China (Chen et al., 2017). Indeed, some studies reported that internally-mixing assumption would overestimate the light-absorbing capability of BC (Cappa et al., 2012;Bond et al., 2013), while others found that the assumption is "relatively reliable for modelling" and can promote the underestimation of modelled $MAC_{BC}$ (Adachi and Buseck, 2008;Koch et al., 2009). Moreover, the simulations of aerosol optical properties and radiative effect conducted over China perform well with internal mixing assumption (Huang et al., 2015;Zhang et al., 2015;Wang et al., 2014a). Thus the results can be seen as an upper limit and is acceptable with the uncertainty of ~20%. Further studies on the representation of the

evolution of mixing state of BC is still needed when more field measurements data in different regions are available.

*2. P5/L17: When the abbreviation regarding "SNA" first appeared in the paper, the author should also give its full terms.*

**Response:** Accepted. We will give the full terms when "SNA" first appears in the revised paper .

*3. Why the author only considered BC mixing with SNA? Why not consider other aerosol compositions, especially organic species that play an important role in the mass fraction of aerosol particles.*

**Response:** Thanks for pointing out this issue. BC was also mixed with organic carbon (OC). There are multiple reasons for excluding OC in our current work. First, the optical properties of organic species such as refractive index and their mixing state with other aerosol components have not yet been fully explored and large uncertainties do exist (Kanakidou et al., 2004). Secondly, scattering aerosols like sulfate and nitrate magnify the light-absorbing capacity of BC not only through their own lensing effect but also through further increase in mass absorption cross section due to hygroscopic growth. Considering the uncertainties in OC's optical properties and less notable hygroscopicity, we do not include it in this article. More quantitatively, we conducted some sensitivity simulation by assuming OC as purely scattering aerosol (default refractive index given in WRF-Chem). As shown in Fig. R1-2, when the inputted organic aerosol surface concentration exceeds 40 $\mu g\ m^{-3}$, the changes in heating rate (~0.02 K $h^{-1}$) and temperature (less than 0.2 $^o$C) are not much different from those in Figure 1-2. However, we do thank the referee for raising this point. We will add a few sentences to explain the reason of excluding OC in our revised manuscript.

[Figure]

Fig. R1 Vertical profile of BC and OC along with shortwave heating rate induced by BC in the afternoon (12:00-16:00). Black dash line denotes the averaged PBL height during this period.

[Figure]

Fig. R2 Diurnal variation of the air temperature change caused by aerosols and of PBL height for runs with (solid line) and without (dash line) ARI. Other settings are the same as Figure 2 except for additional OC input.

*4. Figure 1 shows the heating efficiency of BC due to shortwave radiation absorption peaked around 600-800 m. But the strongest upper-air warming exhibited at 1000-1200 m shown in Fig. 2, different with the height of the largest heating efficiency of BC. The author need to explain the reason.*

**Response:** Although shortwave heating peaked around 600-800 m, due to turbulence

mixing and convective motion, the heated air will not stay where they are. Instead, they will rise because of small density until reaching the warmer capping inversion.

*5. According to the vertical profile of shortwave heating rate included by unit BC mass (Fig. 1), the authors discussed that BC in the upper PBL is more efficient in terms of absorbing shortwave radiation and heating surrounding air masses. To support the author's statement on the upper PBL, Figure 1 need to show the average PBL height in the afternoon (12:00-16:00 LT) during a heavy polluted episode 23rd-24th December, 2013 in Beijing.*

**Response:** Accepted. We re-plotted Figure 1 by adding the daytime averaged (12:00-16:00LT) PBL height during this episode.

*6. P6/L25-27: The authors state here that Figure 3a displayed the information of the incident solar radiation at the surface, decline in surface temperature and surface sensible heat flux. I did not found these information in Fig. 3a. The authors should check.*

**Response:** Accepted. Fig. 3a should be the whole Fig. 3, which gives all information about temperature (Fig. 3b) and radiation (Fig. 3c). Sensible heat flux is directly calculated from model output and is not shown in the figure.

*7. In Fig. 3b, what does the black dash line represent? Authors should state it in the caption of the figure 3.*

**Response:** Accepted. The black dash line represents the diurnal averaged PBL height, and the statement will be added to the revised manuscript.

*8. P7/L4: According to the data shown in Fig.4, the increment of increase BC concentration from 0 to 30 µg m-3 should be 5µg m-3 rather than 2 µg m-3.*

**Response:** Accepted.

*9. P8/L10-15: The calculation method of absorption coefficient amplification factors of BC was different with that in the literature. Authors calculated the absorption amplification of BC aerosols based on the BC/SNA ratio during different periods (i.e., clean period: BC/SNA=1:3, haze episode: BC/SNA=1:8). The obtained absorption amplification (i.e, 1.8, 1.7, 1.6 and 1.4 at wavelength 300, 400, 600 and 1000 nm) was then compared with previous laboratory and in-situ observational studies. However, the absorption amplification of BC aerosols in previous studies represents that the*

*enhancement of light absorption of BC-containing particles due to coating materials on BC surface comparing with bare BC.*

**Response:** Accepted. The calculation of absorption amplification in this study indeed represents the enhancement of BC absorptive properties due to different BC/SNA ratio, or different pollution periods, and should be compared with changes of BC mass absorption cross-section at some specific wavelength. A two-week intensive measurements in the urban area in Northern China winter with both SNA and BC concentrations as well as light absorption coefficient observed are employed here to compare with the simulations (Chen et al., 2017). When BC/SNA ratio rises from 1:3 to 1:8, the light absorption coefficient observed increases by an averaged factor of ~1.5, indicating that the absorption amplification obtained is reasonable. We will add this information in the revised manuscript.

*10. In Fig.5, authors used BC/SNA ratio of 1:3 and 1:8 to calculate the aerosol absorption coefficient at clean period and haze episode, respectively. They should give the data sources of BC/SNA ratio at different periods. Moreover, Fig.5 also shows the solar spectral irradiance at sea level (bright yellow shadow). However, there was not any discussion on this information in the paper. Why did the authors give solar spectral irradiance at sea level in Fig. 5?*

**Response:** The BC/SNA ratio we used in our simulation is extracted from in-situ measurements conducted in Beijing during 1-16 January 2013(Sun et al., 2014). The detailed aerosol data of clean period and haze episode can be found in the Table 1 of this article. In shortwave radiation scheme, only 4 typical bands (300, 400, 600, 1000nm) are calculated. In order to investigate the relative importance of amplification of absorption extinction coefficient for each band, we gave solar spectral irradiance at sea level as a reference, which help the reader to understand our conclusion that the amplification of 400-600 nm is quite important because of more intensive solar irradiance. This discussion will be added to the revised manuscript.

*11. Considering most of days in north China characteristic of RH lower than 50%, I suggest adding a case of RH=30% in Fig.6. Moreover, authors gave the absorption extinction coefficient at a certain wavelength (i.e., 400 nm), they should state it in the caption of the figure 6.*

**Response:** Thanks for the suggestions. In fact, we have conducted the simulation in the case of RH=20%, 30% and 40%. However, the results are the same as RH=50% due to the RH threshold for aerosols' deliquescence. If ambient RH is lower than this threshold, also known as the deliquescence relative humidity (DRH), aerosols will not absorb water and remain solid thus no

change of optical properties occurs. Here we present the result with RH equal to 30% for your reference (Fig. R3). Moreover, the description about the wavelength will be added to the caption in revised manuscript.

[Figure]

Fig. R3 Same as Figure 6 but with RH varying from 30% to 90%

*12. RH at a certain SNA influence not only aerosol particle diameter but also their reflective index (RI). Authors mentioned the change of aerosol particle diameter at different levels of RH (P8/L17-18). Did they also consider the RI change? The authors should give detail description on data process at different levels of RH in Sect. 2 (data and method), such as the diameter growth factors and RI setting in the model calculation.*

**Response:** Yes, RH absolutely modifies not only aerosol diameter but also the RI value. We do include the impact of RH on aerosol RI. Actually, in the WRF-Chem model, the RI value is determined by 11 aerosol species, including aerosol water which is highly dependent on RH (Barnard et al., 2010;Fast et al., 2006). The calculation of extinction coefficient, single-scattering albedo and asymmetry factor are based on a sectional approach. In each bin, the particles are assumed to be spherical and internally mixed. Chemical masses (converted to volumes after) and particle number are required to find physical diameter of each bin. Volume averaged mixing rule are employed to calculate the bulk refractive index of all the particles (including water aerosol) in each bin. The detailed information can be found in Barnard et al. 2010. Therefore, the influence of RH and hygroscopic growth on SNA optical properties technically are already considered in this study. We will add these details in Section 2 in the revised manuscript.

*13. When investigating the impacts of BC aging process on light absorption, authors discussed the difference of absorption under difference BC/SNA ratio, shown in Fig.5*

*and Fig.6. However, why they fixed a value of BC/SNA ratio (i.e., 1:6 shown in Fig. 7) rather than using different BC/SNA ratio as above two figures to further discuss the impacts of BC aging process on PBL height? Figure 5 shows different BC/SNA ratio under different pollution levels (i.e., clean period: BC/SNA=1:3, haze episode: BC/SNA=1:8) based on in-situ observation from literature, indicating that BC/SNA ratio will change with BC concentrations. Therefore, it seems to be unreasonable to assume same BC/SNA ratio under different BC concentrations.*

**Response:** Accepted. There is no need to use a fixed value of BC/SNA ratio since it varies a lot in actual haze episode. Here, according to the comment, we conducted another set of parallel experiments with BC concentration fixed at 5 μg m$^{-3}$ and SNA concentration ranges from 20 to 160 μg m$^{-3}$, which shares the same chemical components with Figure 6. The PBL response of increasing by lower-level BC and decreasing by upper-level BC are both magnified. As the pollution gets severe which characterized by increasing level of scattering aerosols, the amplification of BC induced PBL height changes also increases, as shown in the following figure. We will revise Figure 7 and the relevant discussions.

[Figure]

Fig .R4 Enhanced decreasing/increasing of PBL height (contour map) and enhanced reduction of sensible heat flux at surface (dash isoline) caused by amplified absorption of BC internally mixed with scattering aerosols.

*14. When BC concentration is lower than 2 μg m-3, the PBL height variations due to specific BC distribution shown in Fig. 9 exhibited significant difference with that shown in Fig.4. Why?*

**Response:** After looking through the modelling results, it turns out that the difference is attributed to different criteria of comparison. In Fig. 4, the positive

variations of PBL height is compared to the scenario without aerosol radiation interaction (ARI), while in Fig. 9 the negative variations of the first two layers result from the comparison to the scenario with ARI but no manually inputted aerosols. Therefore, the difference can be viewed as the effect of back ground aerosols. For convenience, the comparison standard will be unified to the difference between the scenarios with/without ARI in the revised version.

**References:**

Adachi, K., and Buseck, P. R.: Internally mixed soot, sulfates, and organic matter in aerosol particles from Mexico City, Atmos. Chem. Phys., 8, 6469-6481, 10.5194/acp-8-6469-2008, 2008.

Barnard, J. C., Fast, J. D., Paredes-Miranda, G., Arnott, W. P., and Laskin, A.: Technical Note: Evaluation of the WRF-Chem "aerosol chemical to aerosol optical properties" module using data from the MILAGRO campaign, Atmospheric Chemistry and Physics Discussions, 10, 8927-8961, 10.5194/acpd-10-8927-2010, 2010.

Bond, T. C., Doherty, S. J., Fahey, D. W., Forster, P. M., Berntsen, T., DeAngelo, B. J., Flanner, M. G., Ghan, S., Kärcher, B., Koch, D., Kinne, S., Kondo, Y., Quinn, P. K., Sarofim, M. C., Schultz, M. G., Schulz, M., Venkataraman, C., Zhang, H., Zhang, S., Bellouin, N., Guttikunda, S. K., Hopke, P. K., Jacobson, M. Z., Kaiser, J. W., Klimont, Z., Lohmann, U., Schwarz, J. P., Shindell, D., Storelvmo, T., Warren, S. G., and Zender, C. S.: Bounding the role of black carbon in the climate system: A scientific assessment, Journal of Geophysical Research: Atmospheres, 118, 5380-5552, 10.1002/jgrd.50171, 2013.

Cappa, C. D., Onasch, T. B., Massoli, P., Worsnop, D. R., Bates, T. S., Cross, E. S., Davidovits, P., Hakala, J., Hayden, K. L., Jobson, B. T., Kolesar, K. R., Lack, D. A., Lerner, B. M., Li, S.-M., Mellon, D., Nuaaman, I., Olfert, J. S., Petäjä, T., Quinn, P. K., Song, C., Subramanian, R., Williams, E. J., and Zaveri, R. A.: Radiative Absorption Enhancements Due to the Mixing State of Atmospheric Black Carbon, Science, 337, 1078-1081, 10.1126/science.1223447, 2012.

Chen, B., Bai, Z., Cui, X., Chen, J., Andersson, A., and Gustafsson, O.: Light absorption enhancement of black carbon from urban haze in Northern China winter, Environ Pollut, 221, 418-426, 10.1016/j.envpol.2016.12.004, 2017.

Fast, J. D., Jr, W. I. G., Easter, R. C., Zaveri, R. A., Barnard, J. C., Chapman, E. G., Grell, G. A., and Peckham, S. E.: Evolution of ozone, particulates, and aerosol direct radiative forcing in the vicinity of Houston using a fully coupled meteorology‐chemistry‐aerosol model, Journal of Geophysical Research Atmospheres, 111, 5173-5182, 2006.

Huang, X.-F., Sun, T.-L., Zeng, L.-W., Yu, G.-H., and Luan, S.-J.: Black carbon aerosol characterization in a coastal city in South China using a single particle soot photometer, Atmospheric Environment, 51, 21-28, 10.1016/j.atmosenv.2012.01.056, 2012.

Huang, X., Song, Y., Zhao, C., Cai, X., Zhang, H., and Zhu, T.: Direct Radiative Effect by Multicomponent Aerosol over China, Journal of Climate, 28, 3472-3495, 10.1175/jcli-d-14-00365.1, 2015.

Kanakidou, M., H. Seinfeld, J., Pandis, S., Barnes, I., Dentener, F., Facchini, M., Van Dingenen, R., Ervens, B., A, N., J. Nielsen, C., Swietlicki, E., Putaud, J.-P., Balkanski, Y., Sandro, F., J, H., Moortgat, G., R, W., Lund Myhre, C., Tsigaridis, K., and Wilson, J.: Organic aerosol and global climate modelling: A review, 2004.

Koch, D., Schulz, M., Kinne, S., McNaughton, C., Spackman, J. R., Balkanski, Y., Bauer, S., Berntsen, T., Bond, T. C., Boucher, O., Chin, M., Clarke, A., De Luca, N., Dentener, F., Diehl, T., Dubovik, O., Easter, R., Fahey, D. W., Feichter, J., Fillmore, D., Freitag, S., Ghan, S., Ginoux, P., Gong, S., Horowitz, L., Iversen, T., Kirkev, aring, g, A., Klimont, Z., Kondo, Y., Krol, M., Liu, X., Miller, R., Montanaro, V., Moteki, N., Myhre, G., Penner, J. E., Perlwitz, J., Pitari, G., Reddy, S., Sahu, L., Sakamoto, H., Schuster, G., Schwarz, J. P., Seland, Ø., Stier, P., Takegawa, N., Takemura, T., Textor, C., van Aardenne, J. A., and Zhao, Y.: Evaluation of black carbon estimations in global aerosol models, Atmospheric Chemistry and Physics, 9, 9001-9026, 10.5194/acp-9-9001-2009, 2009.

Li, W. J., Zhang, D. Z., Shao, L. Y., Zhou, S. Z., and Wang, W. X.: Individual particle analysis of aerosols

collected under haze and non-haze conditions at a high-elevation mountain site in the North China plain, Atmospheric Chemistry and Physics, 11, 11733-11744, 10.5194/acp-11-11733-2011, 2011.

Sun, Y. L., Jiang, Q., Wang, Z. F., Fu, P. Q., Li, J., Yang, T., and Yin, Y.: Investigation of the sources and evolution processes of severe haze pollution in Beijing in January 2013, J Geophys Res-Atmos, 119, 4380-4398, 10.1002/2014JD021641, 2014.

Wang, J., Wang, S., Jiang, J., Ding, A., Zheng, M., Zhao, B., Wong, D. C., Zhou, W., Zheng, G., Wang, L., Pleim, J. E., and Hao, J.: Impact of aerosol–meteorology interactions on fine particle pollution during China's severe haze episode in January 2013, Environmental Research Letters, 9, 094002, 10.1088/1748-9326/9/9/094002, 2014a.

Wang, Q., Huang, R. J., Cao, J., Han, Y., Wang, G., Li, G., Wang, Y., Dai, W., Zhang, R., and Zhou, Y.: Mixing State of Black Carbon Aerosol in a Heavily Polluted Urban Area of China: Implications for Light Absorption Enhancement, Aerosol Science and Technology, 48, 689-697, 10.1080/02786826.2014.917758, 2014b.

Zhang, B., Wang, Y., and Hao, J.: Simulating aerosol–radiation–cloud feedbacks on meteorology and air quality over eastern China under severe haze conditionsin winter, Atmospheric Chemistry and Physics, 15, 2387-2404, 10.5194/acp-15-2387-2015, 2015.

Zhang, G., Bi, X., Li, L., Chan, L. Y., Li, M., Wang, X., Sheng, G., Fu, J., and Zhou, Z.: Mixing state of individual submicron carbon-containing particles during spring and fall seasons in urban Guangzhou, China: a case study, Atmospheric Chemistry and Physics, 13, 4723-4735, 10.5194/acp-13-4723-2013, 2013.

---

## Author Comment (AC2) · 9 Jan 2018

**Response to Referee #2**

*General Comments: This study investigated the black carbon "dome effect" and its key influencing factors, namely the vertical distribution and aging processes of BC, and the underlying land surface. The "dome effect" can play an important role in haze evolutions, which makes this study an interesting topic. Also, the manuscript is well organized and clearly presented, and is worth publishing. However, several concerns need to be addressed before the final publication.*

**Response:** We would like to thank the referees for their time and useful comments towards the improvement of our manuscript. We have made careful considerations and now reply to the comments one by one.

*Specific Comments: 1. One major concern of this study is the lacking of information on actual scenarios. While low-level BC can enhance PBL height while upper-level would suppress that, what is the approximate threshold of low-level BC to upper-level BC concentration ratios, at which these two effects can offset each other? Is this threshold easily reached during haze events? That is, how often and how universal is the "dome effect" present? In actual scenarios, the BC are more likely to be composed of both a low-level freshly emitted peak, and an upper-level transported peak. Their different ratios may lead to different overall effect. Since observation on vertical BC profile is scarce, a relatively long-term simulation covering a larger domain (e.g., northern and eastern China) with actual configurations like the one shown in Fig. 1 might be helpful, or at least this issue should be discussed in more detail.*

**Response:** Thanks for the suggestion. This work is a complementary study to our previous work, Ding et al. (2016), which is based on 3D simulations with actual scenarios. Of course, 3-D modelling for a larger domain and for a longer period will be helpful for quantifying the overall impact of the "dome effect" of BC. However, comprehensive 3D modelling with real scenarios sometimes is difficult to identify the key factors because of its complexity. This is the reason why only 1-D modelling with ideal scenarios was considered in this study. As mentioned, BC vertical distribution can be quite heterogeneous due to synoptic weather conditions or long-distance transport. That is to say, the height of the upper-level transported peak and the ratio of the upper- to lower-level BC concentration can vary a lot (Li et al., 2015;Allen and Landuyt, 2014;Trompetter et al., 2013), and the meteorological conditions in different regions may also lead to varied threshold concentrations. Therefore, it may be difficult to determine a universally applied threshold in different region. Specifically, in Beijing, if the upper-level (about 1000m) and lower-level BC are 5 and 20 μg m$^{-3}$ respectively (Li et al., 2015), which represents the common situation during

heavy polluted days, dome effect will occur with a ratio of upper-level to lower-level BC concentration being about 0.25. Thus in actual scenarios, the threshold of dome effect can be easily reached. We also analysed the result of regional modelling in Ding et al. (2016) for the whole month of December in 2013, and found that dome effect enhanced the atmospheric stability (PBL height decreases over 10%) in 66%, 73%, 69% and 66% of days for four different cities: ZZ (34.73º N, 113.61º E), SJZ (38.04º N, 114.71º E), SH (31.21º N, 121.45º E), NJ (32.08º N, 118.77º E). Furthermore, the frequency increased to over 90% during pollution episode. We also performed our simulations over these cities in northern and eastern China during several winter haze episodes, and we found the same "dome effect" exist in these cities with different PBL decreasing. Hence, the occurrence of "dome effect" can be seen as quite universal and frequent in northern and eastern China when heavy polluted events take place. Relevant descriptions will be added in Section 3.1 in the revised manuscript.

[Figure]

Fig. R1 Diurnal variations of the air temperature change caused by aerosols during haze episode in Zhengzhou (ZZ), Shijiazhuang (SJZ), Shanghai (SH) and Nanjing (NJ), and of PBL height for runs with (solid line) and without (dash line) ARI.

*2. Although the simulation results are well explained, the conclusion about chimneys and domestic stoves seems somewhat abrupt. What is the typical height of chimneys? Can that compare to the height of the inversion layer? It was more confusing on the conclusions about domestic stoves at rural areas. In the context of this manuscript, the*

*depression of PBL at rural areas should be caused mainly by the long-range transported upper-level BC, not the local emitted ones. On the contrary, the freshly emitted BC would serve as the low-level BC and tend to enhance the PBL. Thus the fact that rural areas are more sensitive to "dome effect" would lead to the conclusion that reducing long-range transported upper-level BC is more important. The casual relationship should be better described.*

**Response:** Thanks for raising these points. We agree that the relationship of these discussion should be better described. The typical height of chimneys and stack for power plants are higher than 200 m in China (Hao et al., 2007;Zhou et al., 2003). Further, the emissions of these coal-fired power factories tend to be much warmer than the ambient air and thus will rise and expand to cool down, after which it may possess a height of around 300 m (Ito et al., 2006;Zhou et al., 2003). In a modern coal-fired power plant, the average particle size in the stack gas is often the scale of sub-micrometre to approximately a few micrometres. Hence, the effect of gravitational sedimentation is expected to be quite small. During winter time, nocturnal residual layer could be as low as 200 m, leading to emissions from stacks to be lifted above the inversion layer. Our key point is that the elevated sources are easily to be long-range transported, so the BC reduction from these sources should be paid more attention if the dome effect is considered. As for domestic stoves at rural areas, although this kind of emission sources mainly locate on the ground, daytime convective motions play an important role in lifting near-surface pollutants to upper-level PBL (Wakimoto and L. McElroy, 1986;Gimson, 1997;Li, 2005), where the dome effect of BC will also play important role in enhancing air pollution. Anyway, we will re-organize these sentences to better express our conclusion or suggestions.

*3. Page 2 Line 1: "developed regions like...": change into "the more developed regions like..."*

**Response:** Accepted.

*4. Page 2 Line 6: is the "680 ug/m3" daily average? Later the hourly maximum of ~900 ug/m3 is mentioned, so here need some clarification.*

**Response:** The "680 ug/m$^3$" is the hourly maximum PM$_{2.5}$ concentration for haze pollution in January, 2013 (Wang et al., 2013) , while "~900 ug/m3" is the maximum hourly concentration in December, 2013 (Zheng et al., 2015). These data are from different pollution episode. More description is given in the revised version.

*5. Page 2 Line 17-L18: consider change the expression of "concentration of BC... far more than..."; "more concentration" seems strange.*

**Response:** Accepted. We will change it to "concentration of BC… higher than…" in the revised manuscript.

**Reference:**

Allen, R. J., and Landuyt, W.: The vertical distribution of black carbon in CMIP5 models: Comparison to observations and the importance of convective transport, Journal of Geophysical Research: Atmospheres, 119, 4808-4835, 10.1002/2014jd021595, 2014.

Gimson, N.: Pollution transport by convective clouds in a mesoscale model, 1805-1828 pp., 1997.

Hao, J., Wang, L., Shen, M., Li, L., and Hu, J.: Air quality impacts of power plant emissions in Beijing, Environ Pollut, 147, 401-408, 10.1016/j.envpol.2006.06.013, 2007.

Ito, S., Yokoyama, T., and Asakura, K.: Emissions of mercury and other trace elements from coal-fired power plants in Japan, Sci Total Environ, 368, 397-402, 10.1016/j.scitotenv.2005.09.044, 2006.

Li, J., Fu, Q., Huo, J., Wang, D., Yang, W., Bian, Q., Duan, Y., Zhang, Y., Pan, J., Lin, Y., Huang, K., Bai, Z., Wang, S.-H., Fu, J. S., and Louie, P. K. K.: Tethered balloon-based black carbon profiles within the lower troposphere of Shanghai in the 2013 East China smog, Atmospheric Environment, 123, 327-338, 10.1016/j.atmosenv.2015.08.096, 2015.

Li, Q.: North American pollution outflow and the trapping of convectively lifted pollution by upper-level anticyclone, Journal of Geophysical Research, 110, 10.1029/2004jd005039, 2005.

Trompetter, W. J., Grange, S. K., Davy, P. K., and Ancelet, T.: Vertical and temporal variations of black carbon in New Zealand urban areas during winter, Atmospheric Environment, 75, 179-187, 10.1016/j.atmosenv.2013.04.036, 2013.

Wakimoto, R., and L. McElroy, J.: Lidar Observation of Elevated Pollution Layers over Los Angeles, 1583-1599 pp., 1986.

Wang, Y., Yao, L., Wang, L., Liu, Z., Ji, D., Tang, G., Zhang, J., Sun, Y., Hu, B., and Xin, J.: Mechanism for the formation of the January 2013 heavy haze pollution episode over central and eastern China, Science China Earth Sciences, 57, 14-25, 10.1007/s11430-013-4773-4, 2013.

Zheng, G. J., Duan, F. K., Su, H., Ma, Y. L., Cheng, Y., Zheng, B., Zhang, Q., Huang, T., Kimoto, T., Chang, D., Pöschl, U., Cheng, Y. F., and He, K. B.: Exploring the severe winter haze in Beijing: the impact of synoptic weather, regional transport and heterogeneous reactions, Atmospheric Chemistry and Physics, 15, 2969-2983, 10.5194/acp-15-2969-2015, 2015.

Zhou, Y., Levy, J. I., Hammitt, J. K., and Evans, J. S.: Estimating population exposure to power plant emissions using CALPUFF: a case study in Beijing, China, Atmospheric Environment, 37, 815-826, 10.1016/s1352-2310(02)00937-8, 2003.